# Elevated LAMTOR4 Expression Is Associated with Lethal Prostate Cancer and Its Knockdown Decreases Cell Proliferation, Invasion, and Migration In Vitro

**DOI:** 10.3390/ijms25158100

**Published:** 2024-07-25

**Authors:** Yaser Gamallat, Huseen Alwazan, Rasoul Turko, Vincent Dang, Sima Seyedi, Sunita Ghosh, Tarek A. Bismar

**Affiliations:** 1Department of Pathology and Laboratory Medicine, Cumming School of Medicine, University of Calgary, Calgary, AB T2N 4N1, Canada; yaser.gamallat@ucalgary.ca (Y.G.); rasoul.turko1@ucalgary.ca (R.T.); vincent.dang1@ucalgary.ca (V.D.);; 2Departments of Oncology, Biochemistry and Molecular Biology, Cumming School of Medicine, University of Calgary, Calgary, AB T2N 4N1, Canada; 3Arnie Charbonneau Cancer Institute, Cumming School of Medicine, University of Calgary, Calgary, AB T2N 4N1, Canada; 4Departments of Mathematical and Statistical Sciences and Medical Oncology, Faculty of Medicine and Dentistry, University of Alberta, Edmonton, AB T6G 2R7, Canada; sghosh1@ualberta.ca; 5Tom Baker Cancer Center, Alberta Health Services, Calgary, AB T2N 4N1, Canada; 6Prostate Cancer Centre, Rockyview General Hospital, Calgary, AB T2V 1P9, Canada; 7Alberta Precision Labs, Rockyview General Hospital, Calgary, AB T2V 1P9, Canada

**Keywords:** late endosomal/lysosomal adaptor, MAPK, mTOR activator 4 (LAMTOR4), mammalian target of rapamycin (mTOR), ERG, PTEN

## Abstract

Late endosomal/lysosomal adaptor, MAPK and mTOR, or LAMTOR, is a scaffold protein complex that senses nutrients and integrates growth factor signaling. The role of LAMTOR4 in tumorigenesis is still unknown. However, there is a considerable possibility that LAMTOR4 is directly involved in tumor cell proliferation and metastasis. In the current study, we investigated the protein expression of LAMTOR4 in a cohort of 314 men who were undergoing transurethral resection of prostate (TURP) consisting of incidental, advanced and castration-resistant cases. We also correlated the data with ERG and PTEN genomic status and clinicopathological features including Gleason score and patients’ outcome. Additionally, we performed in vitro experiments utilizing knockdown of LAMTOR4 in prostate cell lines, and we performed mRNA expression assessment using TCGA prostate adenocarcinoma (TCGA-PRAD) to explore the potential differentially expressed genes and pathways associated with LAMTOR4 overexpression in PCa patients. Our data indicate that high LAMTOR4 protein expression was significantly associated with poor overall survival (OS) (HR: 1.44, CI: 1.01–2.05, *p* = 0.047) and unfavorable cause-specific survival (CSS) (HR: 1.71, CI: 1.06–2.77, *p* = 0.028). Additionally, when high LAMTOR4 expression was combined with PTEN-negative cases (score 0), we found significantly poorer OS (HR: 2.22, CI: 1.37–3.59, *p* = 0.001) and CSS (HR: 3.46, CI: 1.86–6.46, *p* < 0.0001). Furthermore, ERG-positive cases with high LAMTOR4 exhibited lower OS (HR: 1.98, CI: 1.18–3.31, *p* = 0.01) and CSS (HR: 2.54, CI: 1.32–4.87, *p* = 0.005). In vitro assessment showed that knockdown of LAMTOR4 decreases PCa cell proliferation, migration, and invasion. Our data further showed that knockdown of LAMTOR4 in the LNCaP cell line significantly dysregulated the β catenin/mTOR pathway and tumorigenesis associated pathways. Inhibiting components of the mTOR pathway, including LAMTOR4, might offer a strategy to inhibit tumor progression and metastasis in prostate cancer.

## 1. Introduction

Prostate cancer (PCa) is one of the most prevalent forms of cancer affecting men. In fact, annually, 1.6 million men are diagnosed with PCa and 366,000 deaths are caused by the disease [1]. Although the Prostate Specific Antigen (PSA) test is a very common diagnostic tool for PCa, false positives and false negatives can still occur [2].

In general, the epidemiology of prostate cancer is complex. Gene mutations are the most common drivers of this cancer [3]. Androgen receptors (*AR*), phosphate and tensin homolog (*PTEN*), and ETS-related gene (*ERG)* are among the most common sites of genomic alteration in PCa. The activation of *AR* leads to the transcriptional regulation of genes involved in cell survival and proliferation [4]. Stimulation of mTOR signaling can affect different pathways through *PTEN* deletion which encourages cell proliferation, migration, and invasion [5]. These changes contribute to the progression of the disease and increase resistance to androgen deprivation therapy [6]. The overexpression of *ERG*, usually resulting from *TMPRSS2-ERG* fusion, is one of the most common genetic alterations in PCa [7]. Several studies indicate that *ERG* overexpression results in more aggressive phenotypes of PCa [7,8,9]. Therefore, identification of prognostic markers that may improve the accuracy of prostate cancer diagnosis and reduce the risk of overdiagnosis is still needed.

The late endosomal/lysosomal adaptor, MAPK and MTOR activator 4 (LAMTOR4) or C7orf59, is a protein belonging to the LAMTOR family and believed to be involved in cell cycle, proliferation, and growth [10]. LAMTOR4 has been reported as a regulator complex which acts as an activator of rapamycin complex 1 (mTORC1) by sensing changes in the levels of amino acids [11]. mTORC1 mediates several cellular processes, such as cell cycle, proliferation, and angiogenesis, which promote the synthesis of critical proteins required for these processes in the cell [12]. Several cancer types, including PCa, show dysregulated mTORC1 and mTORC1 pathways, leading to tumor progression by enhancing cell proliferation, growth, and survival [12]. However, little is known about LAMTOR4 expression in cancer. As no prior studies have examined the involvement of LAMTOR4 in cancer, its role in the disease remains unclear.

In the current study, we are the first to assess LAMTOR4 expression in a cohort of patients with PCa. We assembled tissue microarrays, correlated the results with the clinical outcomes, and performed in vitro experiments to validate the role of LAMTOR4 in PCa.

## 2. Results

### 2.1. High LAMTOR4 Expression Correlates with Poor Overall Survival (OS) and Cancer Specific Survival (CSS)

In our cohort, LAMTOR4 expression was assessed using IHC (Figure 1A). The mean intensity was significantly different between the groups (*p* value 0.005). When investigating the difference between the groups, we noticed that the patients with advanced and castration-resistant disease had worse OS and CSS compared to incidental cases (Figure 1B,C). LAMTOR4 expression (score 3) was more associated with worse outcomes than scores 0, 1 and 2 (negative, weak, and moderate intensity) (Figure 1D,E). Based on this, we assigned score 3 cases as the high-risk group and score 0, 1 and 2 as the low-risk group. High LAMTOR4 expression was significantly associated with poor overall survival (HR: 1.44, CI: 1.01–2.05, *p* = 0.047) and unfavorable CSS (HR: 1.71, CI: 1.06–2.77, *p* = 0.028) (Figure 1 and Table 1).

### 2.2. The Association between High LAMTOR4 Expression and Prognostic Markers PTEN or ERG

Considering the well-established associations of *PTEN* (loss) and *ERG* (gain) with the outcomes and prognosis of prostate cancer patients, we attempted to investigate the prognostic relevance of LAMTOR4 in conjunction with PTEN or ERG. Our cohort showed, as expected, PTEN score 0 (=loss) (Figure 2A,B) or positive ERG (=gain) (Figure 2C,D) were associated with significantly worse outcomes, i.e., both show worse OS and CSS (Figure 2A,B). Additionally, when high LAMTOR expression was combined with PTEN-(score 0) or loss (Figure 2E,F), it showed a significantly stronger association with poorer OS (HR: 2.22, CI: 1.37–3.59, *p* = 0.001) and CSS (HR: 3.46, CI: 1.86–6.46, *p* < 0.0001) compared to LAMTOR4 expression alone. Moreover, high LAMTOR expression combined with ERG positivity also showed augmented association with both OS and CSS ((HR: 1.98, CI: 1.18–3.31, *p* = 0.01) and (HR: 2.54, CI: 1.32–4.87, *p* = 0.005), respectively) (Figure 2G,H and Table 2).

### 2.3. PanCancer Analysis Revealed High Expression of LAMTOR4 mRNA in Most Types of Cancer

PanCancer data analysis of *LAMTOR4* gene expression in 22 types of cancers showed that *LAMTOR4* was significantly overexpressed in 20 types of cancers, including PCa (Figure 3A). TCGA-PRAD data also shows *LAMTOR4* mRNA expression was highly expressed in prostate cancer patients when compared to normal tissues (Figure 3B).

### 2.4. Differential Gene Expression and Gene Signatures Associated with LAMTOR4 Overexpression

The differentially expressed genes associated with increased expression of the *LAMTOR4* gene in the TCGA-PRAD database were obtained and the log fold changes were plotted using a volcano plot as presented in (Figure 3C). Then the top 50 positively correlated and negatively correlated genes were presented as shown by the heatmap. Furthermore, the GSEA of overexpressed genes associated with the overexpression of *LAMTOR4* were mostly involved in enhanced tumorigenesis suggesting involvement of multiple pathways. The expression levels of *LAMTOR4* showed statistically significant positive correlation with tumor cell proliferation (*p* = 0.002), and cellular response to hypoxia as well as DNA repair. Notably, a negative correlation was observed between *LAMTOR4* overexpression and the cell cycle G2M checkpoint pathway (*p* = 0.05), apoptosis (*p* = 0.0001), EMT (*p* = 0.03), and angiogenesis (*p* < 0.0001), as depicted in Figure 4.

### 2.5. LAMTOR4 Expression in Human PCa Cell Lines

LAMTOR4 protein expression was assessed using several PCa cell lines including RWPE1 (normal prostate cells line). Western blot results showed high expression of LAMTOR4 in 22RV1 and LNCaP cell lines (Figure 5A). However, PC3 and DU145 showed relatively moderate expression. RWPE-1 cells showed very low expression. 22RV1, DU-145 and LNCaP were used in the remaining experiments.

### 2.6. LAMTOR4 Knockdown Decreases Proliferation of PCa Cells

The protein expression of the LAMTOR4 post-transfection in 22RV1, LNCaP and DU-145 cells with siRNA had a significantly low expression of LAMTOR4 (Figure 5B). When we performed the colony formation assay, we observed that LAMTOR4 knockdown in DU-145 cells significantly reduced the colony formation assay compared to the negative control (DU-145 cells transfected with scramble siRNA) (Figure 6A). These results indicated that LAMTOR4 has a potential role in PCa tumor cell proliferation.

### 2.7. LAMTOR4 Knockdown Reduces PCa Cell Migration and Invasion of DU-145 Cells

DU-145 cells transfected with siRNA1 had a significantly lower number of migrated cells in the migration assay 48 h post-transfection when compared to the control (DU-145 cells transfected with scramble siRNA) (Figure 6B). Additionally, when we estimated the number of migrated DU-145 cells transfected with siRNA1 or siRNA 2, the number was significantly lower than the number of migrated DU145 cells transfected with scramble siRNA (negative control). The number of migrated DU145 cells transfected with siRNA1 had a *p*-value of 0.0024 and *p*-value of 0.003 when compared with the control. On the other hand, the number of DU145 invasive cells (Figure 6C) was significantly reduced in cells transfected with either LAMTOR4 siRNA1 or siRNA2 compared to the control (scramble siRNA). The number of invasive DU-145 cells transfected with siRNA1, siRNA2 had a *p*-value of 0.0113 and 0.01 when compared to the control, respectively.

### 2.8. LAMTOR4 Knockdown Attenuates Potential Pathways in Tumorigenesis

We further investigated the molecular mechanism involved in the LAMTOR4 knockdown and its relationship with potential tumor pathways. Remarkably, knockdown of LAMTOR4 in LNCaP cell lines shows significant attenuation of β catenin, p GSK, C MYC and p mTOR and upregulation in p PTEN and E Cadherin compared to the negative control (Figure 7). This indicates a potential role of LAMTOR4 in PCa tumorigenesis.

## 3. Discussion

In the current study, we investigated the clinical significance of LAMTOR4 protein expression in a PCa cohort. Our results revealed that LAMTOR4 is significantly overexpressed in prostate cancer tissue compared to adjacent benign tissue. LAMTOR4 is part of the regulator complex involved in the mechanistic mTORC1 signaling pathway. Dysregulation of the mTOR pathway is frequently observed in cancer, including prostate cancer [13]. mTORC1 activity impacts prostate cancer cell growth, proliferation, and metabolism [14]. Our data indicate LAMTOR4 is crucial in cancer development and progression and impacts patients’ clinical outcomes. Higher expression of LAMTOR4 in PCa patients also resulted in a significantly lower overall survival compared to PCa patients with lower expression of the gene.

Interestingly, our study highlights significant prognostic value for LAMTOR4 expression in combination with common genomic alterations such as *PTEN* loss and *ERG* gain, even when adjusted to Gleason Grade Groups, supporting the potential utilization of combined biomarkers for stratifying men with prostate cancer into various risk groups for lethal disease and overall survival. These observations may also explain the added prognostic value of *PTEN* loss and high LAMTOR expression, potentially leading to increased activation of the *PI3K/AKT/mTOR* pathway in prostate cancer [15,16]. Additionally, ERG alterations may also have an impact on mTOR signaling pathway activation in primary prostate cancer and the development of castration-resistant disease [17]. Accumulating evidence shows that LAMTOR4 is an essential protein in mTOR and possibly MAPK pathways, but further investigation is needed. The mTOR pathway plays an important role in proper cell homeostasis and significantly controls cell survival. This is possibly caused by its function as a scaffold protein and its participation in different cell proliferation pathways [18,19]. These findings suggest that LAMTOR4 is likely involved in tumorigenesis.

We further explored the PanCancer database to investigate *LAMTOR4* gene expression in 22 types of cancers in tumor and normal tissues. *LAMTOR4* expression was significantly increased in all types of cancers compared to normal tissues, except AML, skin, and stomach cancer. Recent data indicates that *LAMTOR4* controls mTOR signaling integrity and tumorigenesis [20]. This is in line with our observations, as the *LAMTOR4* gene potentially functions as an oncogene and its expression may play a role in tumor cell proliferation and metastasis.

To further explore the mechanism underlying *LAMTOR4* expression and PCa progression, we performed gene set enrichment analysis of differentially expressed genes associated with *LAMTOR4* overexpression in PCa by using the TCGA-PRAD database. Our data revealed that overexpression of the *LAMTOR4* gene was generally associated with enhanced tumorigenicity processes in PCa. For example, there was a positive correlation with tumor proliferation signature genes, and a negative correlation with apoptosis genes due to the overexpression of *LAMTOR4*.

To further investigate the role of LAMTOR4 in PCa in vitro, we screened the protein expression of LAMTOR4 using several human PCa cell lines to identify the target cell lines expressing the LAMTOR4 protein. Interestingly, we found that most PCa cell lines such as LNCaP, 22RV1, PC3 and DU145 showed moderate to high expression of the gene, but low expression in normal prostate cell line RWPE1. The knockdown of LAMTOR4 in DU145 cells significantly reduces cell proliferation, as we saw a significant reduction in the number of colonies in the colony formation assay compared to the control. We also investigated the in vitro role of LAMTOR4 in metastasis by using the migration and invasion assay. LAMTOR4 gene knockdown in DU145 cells significantly reduced both the migratory and invasive capabilities of cells compared to the control. These findings confirm that LAMTOR4 plays a role in the metastasis of human PCa cell lines.

In prostate cancer, the knockdown of LAMTOR4 resulted in downregulation of beta-catenin, a key player in cell adhesion and gene transcription, and the upregulation of phospho-PTEN, a phosphorylated form of the tumor suppressor PTEN observed in our experiments. The reduction in beta-catenin levels can impair cell proliferation and survival pathways often activated in cancer cells [21]. Concurrently, the increase in phospho-PTEN suggests enhanced PTEN activity, which negatively regulates the PI3K/Akt/mTOR pathway, thereby inhibiting mTOR signaling and potentially reducing tumor growth and progression [13,22]. This dual effect indicates a potential mechanism by which LAMTOR4 knockdown exerts anti-tumorigenic effects through the modulation of these critical molecular pathways.

The data presented here supports an important and vital role for LAMTOR4 as a biomarker associated with prostate cancer progression and metastasis and with significant clinical outcome prognostic value as an individual biomarker or in association with *PTEN* loss/*ERG* expression.

## 4. Materials and Methods 

### 4.1. Study Population and Tissue Microarray Construction

We assessed the protein expression of LAMTOR4 in a cohort consisting of 314 men diagnosed with PCa who were undergoing transurethral resection of prostate (TURP). The samples consist of incidental (n = 92, 29.3%), advanced (n = 96, 30.6%), and castration-resistant prostate cancer (CRPC) (n = 126, 40.1%). Usually, a well-differentiated cancer emerging in the transition zone and incidentally discovered within TURP chips is termed an “incidental” cancer. All the patients’ demographics are in Table 3.

All specimens were confirmed by the study pathologists to verify the histological diagnosis on initial slides. Gleason scoring was assessed according to the 2014 World Health Organization/International Society of Urological Pathology GGs (ISUP) criteria [23]. Clinical follow-up information was recorded from the Alberta Tumor Registry for dates of therapy, overall survival (OS), and prostate cancer specific mortality (PCSM). The cohort’s samples were assembled onto two tissue microarrays (TMAs) with an average two cores per patient using a manual tissue arrayer (Beecher Instruments, Silver Spring, MD, USA). The study was approved by the institutional review board of the Cumming School of Medicine, University of Calgary, Calgary, Alberta, Canada.

### 4.2. Immunohistochemistry

Protein expression of LAMTOR4, ERG and PTEN were assessed using a Dako Omnis auto-stainer (Dako North America Inc., Carpinteria, CA, USA). Briefly, 4 µm formalin-fixed paraffin-embedded sections (FFPE) were deparaffinized, hydrated and pretreated with Dako epitope retrieval buffer pH 9.0. rabbit polyclonal LAMTOR4 at (1:100) dilution (Cell Signaling, Danvers, MA, USA), rabbit monoclonal ERG antibody at (1:100) dilution (Clone EPR 3864, Epitomics, Burlingame, CA, USA) and anti-PTEN rabbit monoclonal (138G6, Cell Signaling Technology, Danvers, MA, USA) at 1:25 dilution. The FLEX DAB+ Substrate Chromogen system was used as the post-incubation detection reagent.

### 4.3. Pathological Analysis

The study pathologists confirmed histological diagnoses for each TMA core. Gleason scoring was assessed. Both predominant patterns of prostate cancer per patient were sampled and incorporated into the TMAs for analysis. Figure 1A depicts the protein expression of tissues, distinguishing between LAMTOR4-positive and -negative samples.

LAMTOR4 and PTEN IHC expression were assessed using a four-tiered system (0, negative; 1, weak; 2, moderate; and 3, high expression). We further categorized the expression to high and low risk-based groups for statistical analysis: LAMTOR low risk (negative; score 0,1 and 2) and high risk (score 3); PTEN high risk (score 0; negative) and low risk (weak, moderate, and high; score 1–3). ERG was categorized as negative (score 0) or positive (score 1) based on correlation to ERG gene rearrangement as detected by FISH.

### 4.4. Cell Lines and Cell Culture

All cell lines used in this study were purchased from American Type Culture Collection (ATCC; Manassas, CA, USA) and maintained in our lab. DU-145, LNCaP, and 22RV1 cell lines were sub-cultured and maintained in RPMI 1640 medium (GIBCO life technology, Grand Island, NY, USA). PC3, PC3-Luc, and PC3-ERG cells were cultured in DMEM/F12 medium (GIBCO life technology, Grand Island, NY, USA). All media were supplemented with 10% fetal bovine serum (FBS) (GIBCO life technology, Grand Island, NY, USA) and 1% Penicillin-Streptomycin (GIBCO life technology, Grand Island, NY, USA). RWPE-1 cells were cultured in Keratinocyte-Serum Free Medium supplemented with growth factors (GIBCO life technology, Grand Island, NY, USA). All cells were incubated at 37 °C in 5% CO_2_ environment.

### 4.5. Cell Line Transfection and RNA Silencing

To knock down the *LAMTOR4* gene in PCa cell lines, a pre-designed LAMTOR4 Silencer Select short interference RNA (siRNA) was purchased from (Ambion, Grand Island, NY, USA), and a scramble siRNA served as a negative control. As directed by the manufacturer, PCa cell lines (22RV1, DU145 and LNCaP) were seeded in a 6 well plate and were grown until they reached 70% to 90% confluency. The cells were washed with DPBS (GIBCO life technology, Grand Island, NY, USA). The transfection reaction premix was prepared in Opti-MEM (GIBCO life technology, Grand Island, NY, USA) by adding the siRNA to create the siRNA transfection mixture and Lipofectamine RNAiMAX (Invitrogen, Carlsbad, CA, USA). The siRNA reaction mix was added to each well, and the cells were incubated at 37 °C in 5% CO_2_ environment. The knockdown efficiency of siRNA on the LAMTOR4 gene was checked using Western blot at 24 h interval up to 96 h.

### 4.6. Western Blot

Total proteins from cell lines were extracted using radioimmunoprecipitation assay (RIPA) lysis buffer (Cell Signaling, Danvers, MA, USA) containing Protease/Phosphatase inhibitors (Cell Signaling, Danvers, MA, USA). After denaturation with sample buffer (Sigma-Aldrich, St. Louis, MO, USA), about 20 µg of each lysate was loaded into polyacrylamide SDS-PAGE gel to separate proteins based on their molecular weight. The proteins were then transferred from the gel into a Polyvinylidene difluoride (PVDF) or Nitrocellulose (NC) membrane (BIO-RAD Immun-Blot^®^ Membrane, Bio-Rad Laboratories, Hercules, CA, USA). The membrane was then blocked in Tris-buffered saline (TBS) containing 5% skimmed milk for 1 h at room temperature or EveryBlot Blocking Buffer (Bio-Rad Laboratories, Hercules, CA, USA) for 5 min at RT. The membrane was then incubated with the primary antibody overnight at 4 °C with continuous shaking. The membrane was then washed 3 times with TBST (5 min each wash), and then incubated with either anti-rabbit IgG or anti-mouse IgG secondary antibody combined to HRP horseradish peroxidase (Cell Signaling, Danvers, MA, USA) for 1 h while shaking at RT. After washing, the membrane exposure was performed using ECL Chemiluminescence (Bio-Rad Laboratories, Hercules, CA, USA) and signal detected by using the ChemiDoc imaging system (Bio-Rad Laboratories, Hercules, CA, USA).

LAMTOR4 protein expression was assessed using the human PCa cell lines (LNCaP, 22RV1, DU-145, PC3, and RWPE-1) by Western blot technique, while 22RV1, DU145 and LNCaP were used for the remaining experiments.

### 4.7. Colony Formation Assay

The colony formation assay was performed to assess the potential role of LAMTOR4 on cell proliferation. Briefly, DU145 cells were transfected using siRNA for LAMTOR4 or scramble siRNA (negative control) as described above. After 24 h, live cells were stained using trypan blue staining method (Sigma-Aldrich, St. Louis, MO, USA) and counted using an Olympus automatic cell counter (Olympus). About 1000 viable cells were seeded in a 6 well plate that contains media with 1% FBS. The plate was then incubated at 37 °C in 5% CO_2_ atmosphere for 10 days to allow cells to grow and proliferate. After that, the cells were fixed and stained with diff-Quick solution. The wells were then washed with ddH_2_O water, and the number of colonies for each treatment were counted using a stereomicroscope.

### 4.8. Migration and Invasion Assay

The migration and invasion assay were used to assess the effect of LAMTOR4 knockdown on PCa cell metastasis by measuring the migratory and invasive capabilities of those cells. DU145 cells were first seeded into a 6 well plate and knocked down as described above. The transfected cells were left to incubate for 24 h at 37 °C in 5% CO_2_ environment. The cells were then trypsinated and counted using an automatic cell counter (Olympus, Breinigsville, PA, USA). Approximately 25,000 of the cells of each treatment were placed into a 0.8 µ insert Corning BioCoat control inserts (Ref# 354578, Corning, Bedford, MA, USA) for the migration assay, while the same number of cells were placed into a Corning Matrigel invasion chamber (Ref# 354480, Corning, Bedford, MA, USA) for the invasion assay. After 24 h, the cells were fixed and then stained with Diff Quick (Siemens Healthcare diagnostics, Tarrytown, NY, USA) to prepare the cells for counting. Magnifications of 10× and 4× were used to take images of the cells by using an inverted EVOS FL life microscope. Several frames were used to count the number of cells and they were averaged.

### 4.9. LAMTOR4 Expression in the Cancer Genome Atlas TCGA-PRAD

The TCGA-PRAD transcriptomics database was used to investigate the potential role of the *LAMTOR4* gene in the PRAD public cohort [24]. Initially, we explored the expression of the *LAMTOR4* gene across 22 different types of tissues, including PCa, by the Pan-cancer analysis tool, using the TCGA, GTex, and TARGET databases. Moreover, LinkedOmics (http://www.linkedomics.org) (accessed on 12 December 2023) was used alongside the TCGA-PRAD database to obtain the differentially expressed gene list associated with *LAMTOR4* overexpression [25]. Finally, we performed analysis of *LAMTOR4* gene expression and signature correlations in TCGA-PRAD.

We conducted a comprehensive analysis utilizing scatter plots, through R, to assess the relationships between *LAMTOR4* gene expression levels and various biological pathway signatures. Each scatter plot represents a distinct dataset with individual data points mapped onto an *x* and *y* axis. Uniformity in design was maintained across all plots, with the *x*-axis uniformly representing the log2-transformed and normalized expression levels, expressed as transcripts per million (TPM + 1), derived from RNA sequencing data. The figures represent the expression levels of the *LAMTOR4* gene, as labeled on the axes. The *y*-axis in each plot refers to a different variable, which changes from one plot to another, covering a variety of biological pathway signatures and processes relevant to our research.

### 4.10. Statistical Analysis

Descriptive statistics were reported for the study variables. Mean and standard deviations were reported for continuous variables. Frequencies and proportions were reported for categorical variables. Independent t-test was used to compare the mean values between two groups, and Chi-square tests were used to compare the proportions between two categorical variables. Overall survival (OS) and prostate cancer-specific mortality (PCSM) were analyzed using Kaplan-Meier estimates. OS was calculated from date of diagnosis to death due to any cause, and patients who were alive at the last date of follow-up were censored. PCSM event was defined as death due to prostate cancer, while patients remaining alive or who died due to other reasons were censored. Log rank statistics were used to compare the KM curves between the groups. Adjusted and unadjusted analysis of OS and PCSM were conducted using Cox’s proportional hazard model. OS and PCSM models were adjusted for Gleason score. Hazard ratio (HR) and the corresponding 95% confidence intervals were reported. A *p*-value < 0.05 was used for all statistical significance. SPSS version 29 (IBM Corp. (2021). IBM SPSS Statistics for Windows. Armonk, NY, USA: IBM Corp.) was used to conduct all statistical analysis.

## Figures and Tables

**Figure 1 ijms-25-08100-f001:**
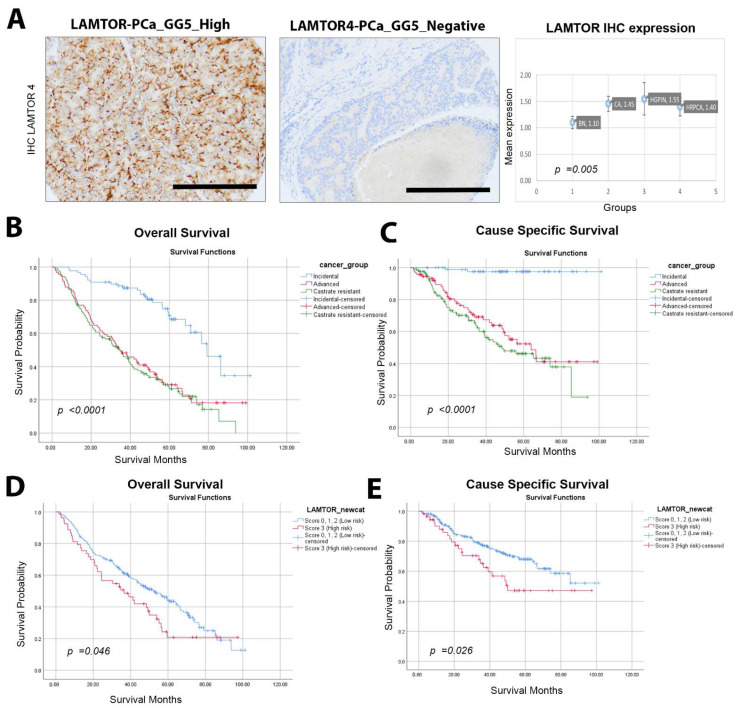
LAMTOR4 expression in PCa and associated clinical outcomes. (**A**) Immunohistochemistry staining of LAMTOR4 in prostate tissues showing high and low expression patterns (scale bar = 100 μm) and the mean expression of LAMTOR4 in Benign (BN), Cancer (Ca), High-grade prostatic intraepithelial neoplasia (HGPIN) and HRPIN tissue samples. The expression levels were scored through IHC. Each sample was scored semi-quantitatively using a four-tiered system (negative—0; weak—1; moderate—2; strong—3). The error bars indicate the standard error of the mean. Statistical analysis was performed using ANOVA test. (**B**) Kaplan–Meier curve showing the overall survival of PCa patients relative to LAMTOR4 protein expression levels when grouped as incidental, advanced and castration-resistant. (**C**) Cause-specific survival. (**D**) KM curve showing the overall survival of PCa patients in relation to LAMTOR4 protein expression levels. (**E**) KM curve for LAMTOR4 cause-specific survival.

**Figure 2 ijms-25-08100-f002:**
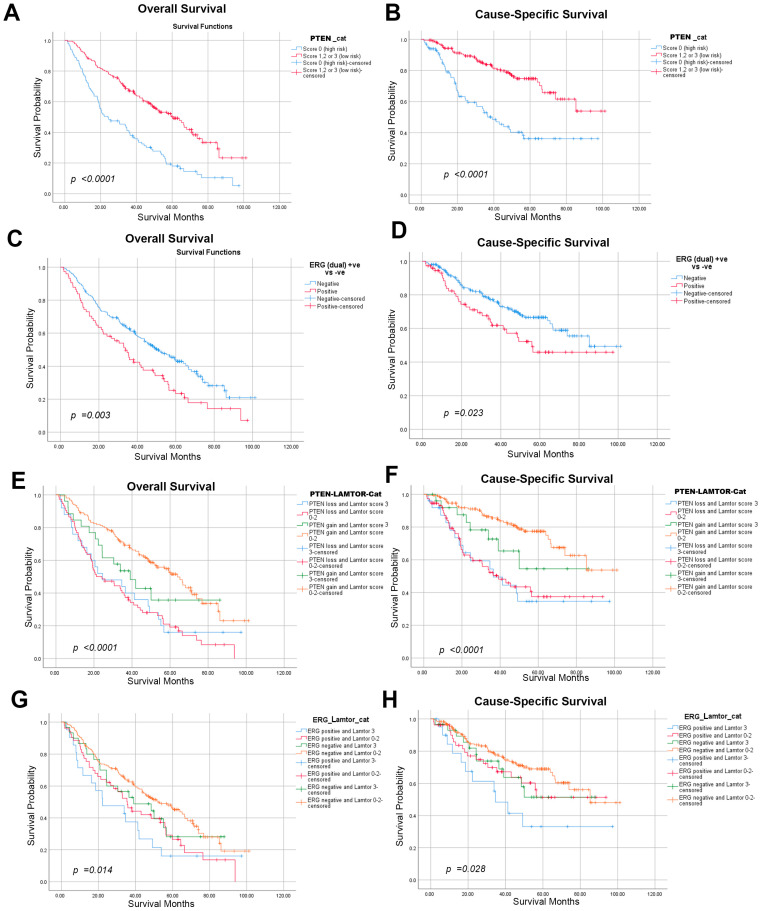
The association between LAMTOR4 expression and PTEN or ERG. KM curves representing the survival of our cohort cases. (**A**,**B**) The PTEN expression patterns and associated OS and CSS, (**C**,**D**) The ERG expression patterns and associated OS and CSS. (**E**,**F**) The combination between LAMTOR4 and PTEN expression in association with OS and CSS. (**G**,**H**) The combination between LAMTOR4 and ERG expression in association with OS and CSS. LAMTOR4 low risk score: (0; negative, 1; weak, 2; moderate), High Risk score 3. PTEN score: 0; negative, score 1, 2, 3; positive. ERG score: 0; negative, score 1; positive.

**Figure 3 ijms-25-08100-f003:**
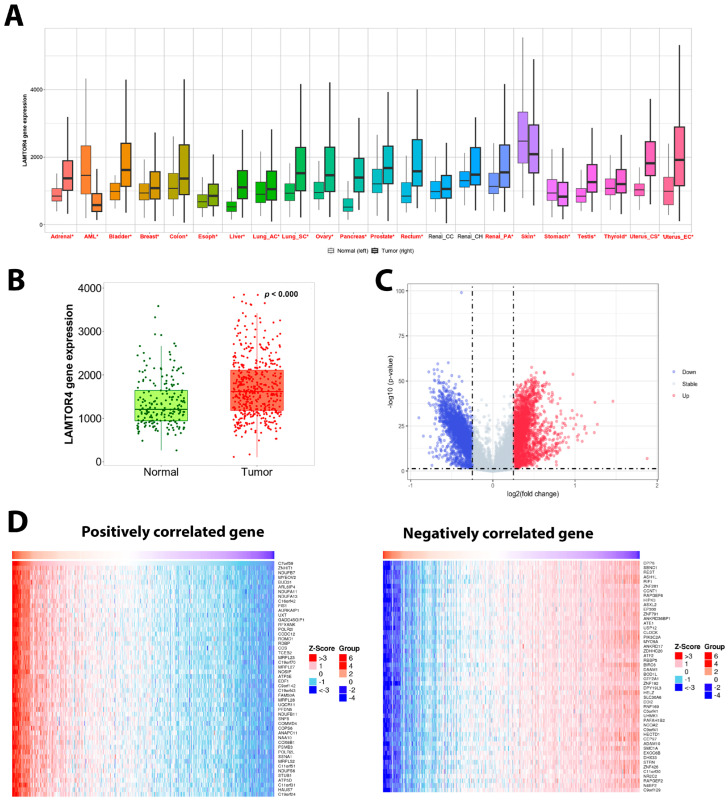
LAMTOR4 mRNA expression in Pan-Cancer data and TCGA PRAD analysis in normal and tumor tissues. (**A**) Box plot showing the mRNA expression levels of *LAMTOR4* in normal vs. tumor tissues of 22 types of cancers. *LAMTOR4* expression was significantly increased in tumor tissues compared to normal tissues in all types of cancers, except in AML, skin, stomach and renal cancer tissues (cancers with significant difference shown in red). * Indicated significant difference *p* < 0.05. (**B**) Box plot showing the mRNA expression levels of *LAMTOR4* in tumor compared to normal prostate tissues. (**C**) Volcano plot showing altered genes associated with LAMTOR4 overexpression. (**D**) Heat map showing the top 50 genes positively and negatively correlated with LAMTOR4 high expression. The blue color represents downregulated genes, the white color represents unchanged genes, and the red color represents gene upregulation.

**Figure 4 ijms-25-08100-f004:**
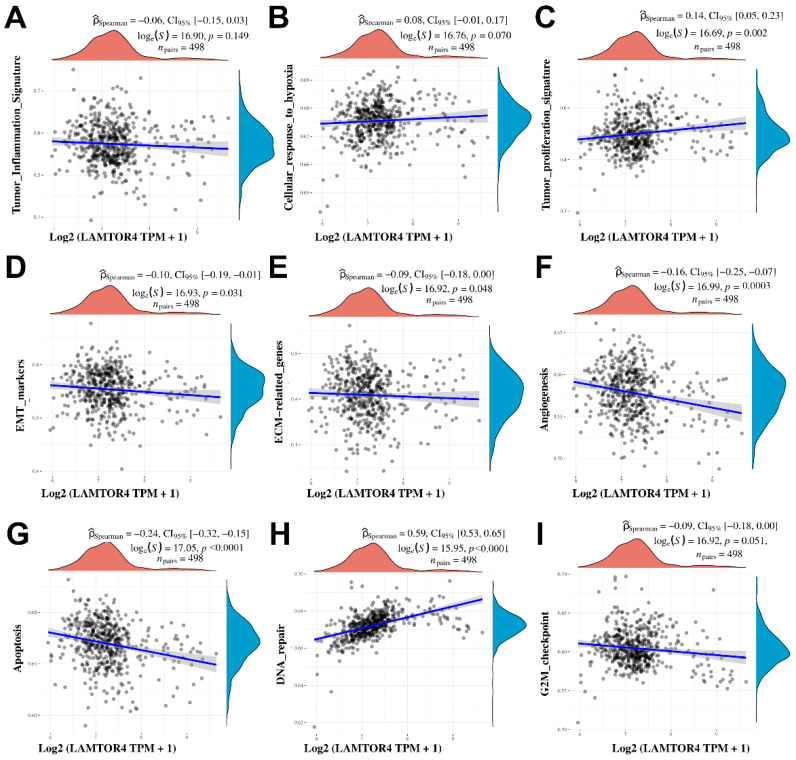
Gene signature correlation analysis of overexpressed *LAMTOR4* in TCGA-PRAD. Scatters blot showing the correlation between mRNA expression and (**A**) Tumor inflammation signature. (**B**) Cellular response to hypoxia. (**C**) Tumor proliferation signature. (**D**) EMT markers. (**E**) ECM-related genes. (**F**) Angiogenesis. (**G**) Apoptosis. (**H**) DNA repair. (**I**) Cell cycle G2M checkpoints. *LAMTOR4* expression in log2 transcript per million (TPM). Correlation analysis performed using Spearman test. Each scatter plot derived from RNA sequencing data represents a distinct dataset with individual data points mapped onto an *x* and *y* axis. Uniformity in design was maintained across all plots, with the *x*-axis uniformly representing the log2-transformed and normalized expression levels, expressed as transcripts per million (TPM + 1).

**Figure 5 ijms-25-08100-f005:**
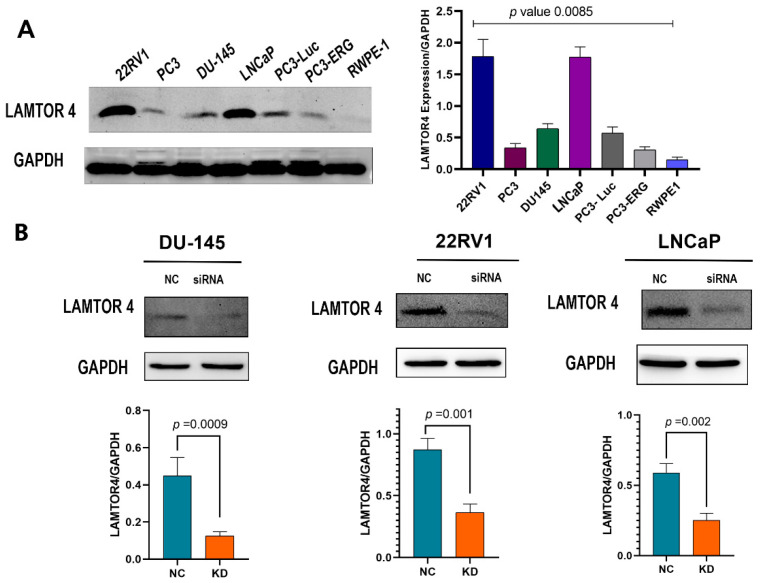
The expression of LAMTOR4 protein in several human prostate cancer cell lines and the knockdown efficiency. (**A**) Western blot analysis of protein expression of LAMTOR4 and GAPDH (control) of several PCa cell lines (22RV1, LNCaP, PC3, DU-145, PC3-luc, PC3-ERG and RWPE-1). GAPDH was used as loading control for data normalization. Bar plots represent relative expression. One-way *ANOVA* was used to analyze the significance differences. (**B**) Western blot analysis in DU-145, 22RV1 and LNCaP cell line post transfection with either siRNA-LAMTOR4, or scramble siRNA (negative control) after 48 h. Bar plots. Significance difference analyzed using *t* test.

**Figure 6 ijms-25-08100-f006:**
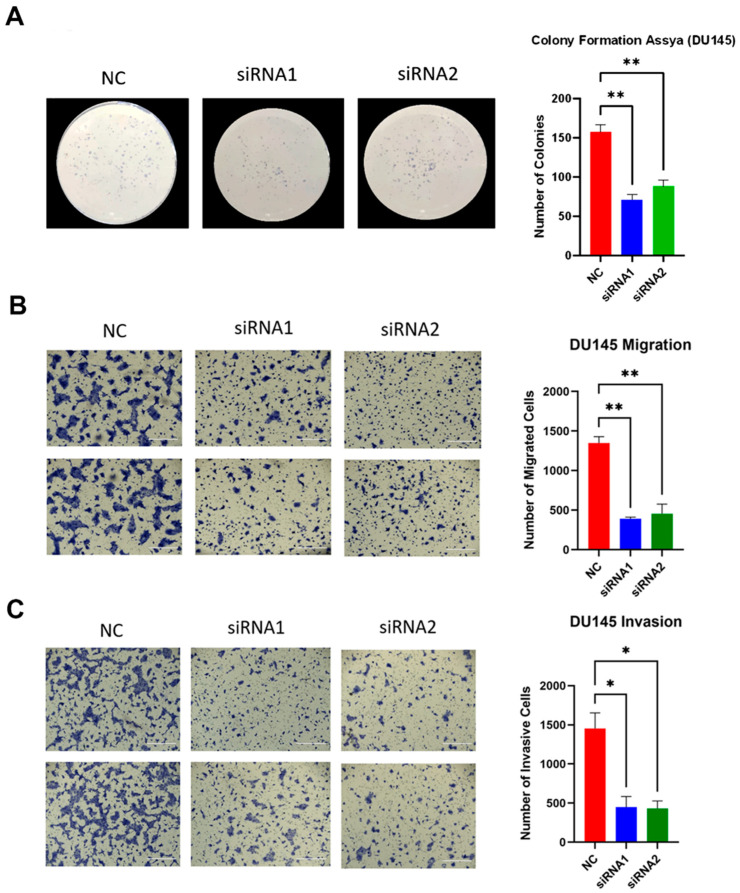
Knockdown of LAMTOR4 decreased DU-145 cells proliferation, migration and invasion. (**A**) Colony formation assay showing DU-145 cell colonies formed after two weeks from transfecting cells with either siRNA1, siRNA2, or scramble siRNA (negative control). Bar plots represent analysis of the colony formation assay using DU-145 two weeks post-transfection. ** *p*-values (<0.01). (**B**) Transwell assay showing the migrated DU-145 cells (in blue) after 48 h post-transfected with either siRNA1, siRNA2, or scramble siRNA as a negative control. The scale bar is 400 µM. Bar plots represent data analysis of migration assay of transfected DU145 cells 48 h post-transfection. Asterisk ** indicates *p*-value < 0.001 when compared with the control. (**C**) Transwell assay using Matrigel invasion assay method 48 h post transfecting cells with either siRNA1, siRNA2, or scramble siRNA as a negative control. The scale bar is 400 µM. Bar plots represent analysis of invasion cells; * indicates *p*-value < 0.05 when compared with the control.

**Figure 7 ijms-25-08100-f007:**
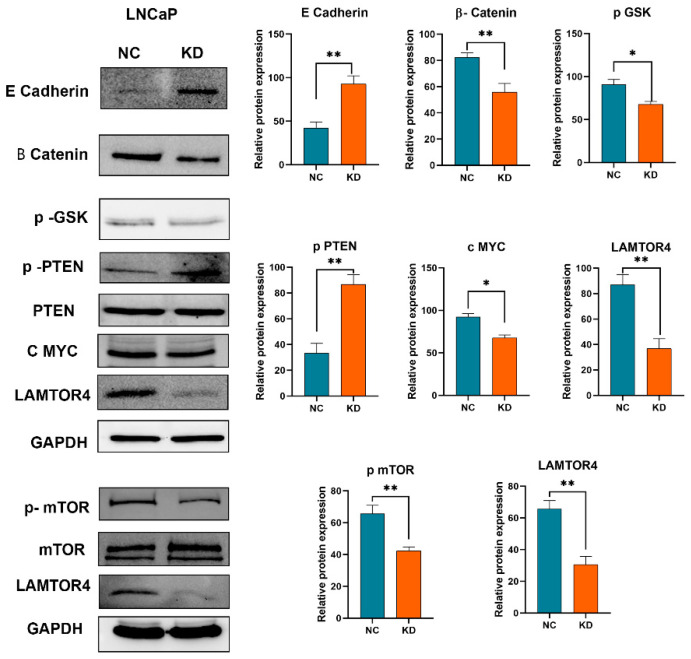
LAMTOR4 Knockdown modulates PCa tumorigenesis through MAPK/mTOR. Western blot data of E Cadherin, β Catenin, p GSK, p PTEN, C MYC, LAMTOR4, p mTOR proteins expression, GAPDH used as loading control. Boxplots represent t-test statistical analysis between the NC and KD. Asterisks * for *p*-value < 0.05, ** for *p*-value < 0.01.

**Table 1 ijms-25-08100-t001:** Association of LAMTOR expression to Gleason Grade grouping, PTEN and ERG status.

Variables	LAMTOR4 Low Risk (Score 0, 1, 2)	LAMTOR4 High Risk (Score 3)	*p*-Value
Gleason Score			
≤6	94 (37.6)	11 (22.0)	0.265
3 + 4	15 (6.0)	5 (10.0)	
4 + 3	24 (9.6)	6 (12.0)	
8	16 (6.4)	5 (10.0)	
9–10	101 (40.4)	23 (46.0)	
PTEN Intensity			
Score 0 (negative)	78 (32.8)	25 (48.1)	0.037
Score (1, 2, 3) (positive)	160 (67.2)	27 (51.9)	
ERG Expression			
Negative	183 (76.9)	31 (59.6)	0.01
Positive	55 (23.1)	21 (40.4)	

**Table 2 ijms-25-08100-t002:** Assessment of individual and combined biomarkers and Gleason Grade grouping with OS and CSS in univariate and multivariate analysis, adjusted for Gleason Grade grouping.

Variables	Overall Survival HR (95% CI)	*p*-Value	Cause-Specific Survival HR (95% CI)	*p*-Value
PTEN (Positive- score 1, 2 or 3)				
Negative (score 0)	2.29 (1.70–3.08)	<0.0001	3.10 (2.05–4.68)	<0.0001
ERG (Negative)				
Positive	1.61 (1.18–2.22)	0.003	1.65 (1.07–2.55)	0.025
GG 1				
GG2	2.73 (1.145–5.13)	0.002	12.18 (3.05–48.73)	<0.0001
GG3	2.21 (1.24–3.95)	0.008	3.91 (0.77–18.88)	0.102
GG4	3.59 (1.87–6.90)	<0.0001	18.22 (4.70–70.68)	<0.0001
GG5	4.63 (3.08 -6.95)	<0.0001	30.95 (9.71–98.68)	<0.0001
LAMTOR (Low risk Score 0, 1, 2)				
LAMTOR- high risk score 3	1.44 (1.01–2.05)	0.047	1.71 (1.06–2.77)	0.028
Combination PTEN and LAMTOR (PTEN-score 1, 2, 3 and LAMTOR-score 0, 1, 2)				
PTEN-score 0 and LAMTOR-score 3	2.22 (1.37–3.59)	0.001	3.46 (1.86–6.46)	<0.0001
PTEN-score 0 and LAMTOR-score 0–2	2.50 (1.79–3.50)	<0.0001	3.36 (2.10–5.38)	<0.0001
PTEN-score 1, 2, 3 and LAMTOR-score 3	1.52 (0.87–2.64)	0.14	1.80 (0.83–3.89)	0.137
Combination PTEN and LAMTOR (PTEN-score 1, 2, 3 and LAMTOR-score 0, 1, 2) *				
PTEN-score 0 and LAMTOR-score 3	1.15 (0.67–1.99)	0.61	1.23 (0.59–2.52)	0.582
PTEN-score 0 and LAMTOR-score 0–2	1.75 (1.23–2.49)	0.002	1.92 (1.17 -3.17)	0.01
PTEN-score 1, 2, 3 and LAMTOR-score 3	1.55 (0.88–2.72)	0.126	2.07 (0.94–4.55)	0.071
Combination ERG and LAMTOR (ERG- negative and LAMTOR- score 0–2)				
ERG positive and LAMTOR score 3	1.98 (1.18–3.31)	0.01	2.54 (1.32–4.87)	0.005
ERG positive and LAMTOR score 0–2	1.57 (1.09–2.26)	0.017	1.47 (0.87- 2.49)	0.152
ERG negative and LAMTOR score 3	1.26 (0.77–2.06)	0.357	1.40 (0.73–2.69)	0.313
Combination ERG and LAMTOR (ERG- negative and LAMTOR- score 0–2) *				
ERG positive and LAMTOR score 3	1.13 (0.64–2.01)	0.67	1.08 (0.51–2.31)	0.834
ERG positive and LAMTOR score 0–2	1.10 (0.75–1.62)	0.625	0.87 (0.50–1.51)	0.615
ERG negative and LAMTOR score 3	1.05 (0.63–1.74)	0.859	1.07 (0.54–2.13)	0.84

* Adjusted for Gleason score, LAMTOR4 score: 0; negative, 1; weak, 2; moderate and 3; high. PTEN score: 0; negative, score 1, 2, 3; positive. ERG score: 0; negative, score 1; positive.

**Table 3 ijms-25-08100-t003:** Patient demographics and descriptive data.

Variables	LAMTOR4 (Score 0, 1, 2)	LAMTOR4 (Score 3)	*p*-Value	Total
Age at diagnosis (years)	75.99 (9.28)	78.39 (8.67)	0.081	76.40 (9.21)
Gleason Score			0.084	
GS 6 or 7	82 (41.2)	10 (26.3)		92 (38.8)
GS 8, 9 or 10	117 (58.8)	28 (73.7)		145 (61.2)
Cancer subgroup			0.015	
Incidental	82 (31.5)	10 (18.5)		92 (29.3)
Advanced	83 (31.9)	13 (24.1)		96 (30.6)
Castration-resistant	95 (36.5)	31 (57.4)		126 (40.1)
Metastasis			0.112	
Yes	28 (10.8)	10 (18.5)		38 (12.1)
No	232 (89.2)	44 (81.5)		276 (87.9)
Hormone therapy			0.016	
Prior to surgery	42 (40.4)	19 (65.5)		72 (54.1)
After surgery	62 (59.6)	10 (34.5)		72 (54.1)
Radiation Therapy			0.743	
Prior to surgery	18 (29.5)	7 (33.3)		25 (30.5)
After surgery	43 (70.5)	14 (66.7)		57 (69.5)
Chemotherapy			0.999	
Prior to surgery	5 (12.9)	1 (9.1)		5 (11.9)
After surgery	27 (87.1)	10 (90.9)		37 (88.1)
Deceased			0.068	
Yes	147 (57.0)	38 (70.4)		185 (59.3)
No	111 (43.0)	16 (29.6)		127 (40.7)
Specific cause of death			0.565	
Prostate cancer-specific death	70 (27.1)	22 (40.7)		92 (29.5)
Other causes or alive	188 (72.9)	32 (59.3)		220 (70.5)
Median follow-up (months)	43 (1.5–101)	34 (2.3–97.3)	0.091	39.0 (1.5–101.2)

## Data Availability

The TCGA PRAD data are available under dbGaP Study Accession phs000178.

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
