# Peer review of "Elevated LAMTOR4 Expression Is Associated with Lethal Prostate Cancer and Its Knockdown Decreases Cell Proliferation, Invasion, and Migration In Vitro"

_ijms, 2024, doi:10.3390/ijms25158100_

Round 1

Reviewer 1 Report (New Reviewer)

Comments and Suggestions for Authors

The manuscript answers the question that LAMTOR4 is involved in PCa cell proliferation and metastasis. The authors investigated the clinical significance of LAMTOR4 protein expression in PCa cohort. They correlated the data with ERG and PTEN status and clinic-pathological features, Gleason score, patients outcome. High LAMTOR4 protein expression was significantly associated with poor OS and CSS. In vitro experiments shown knockdown of LAMTOR4 decrease PCa cells proliferation, migration, and invasion. LAMTOR4 might be as a prognostic marker that may improve the accuracy of prostate cancer diagnosis.

Overall, the paper is well organized and its presentation is clear. The following are the questions and some mistakes in this manuscript:

1)  There are many description errors in the manuscript. Cell names are not written properly. In line 155 and 257 page 4, cell name is LNCaP. While in line 302 page 11, Figure 4, line 311, 314 page 12, line 347 page 14, Figure 6, cell name is LnCAP. In line 297 page 11, B should be H. In line 302 page 11, Fig. 4A should be Fig. 5A. In line 323 page 12, Fig. 5A should be Fig. 5B. There are many mistakes like these. Please check and correct.

2)  In Figure 3A,the authors mentioned LAMTOR4 expression was significantly increased in tomor tissues compared to normal tissues in all types of cancers, except in renal tissues. But LAMTOR4 expression was decreased in AML, Skin, and Stomach cancers.

3)  In Figure 4, the captions A-I are too small and unclear. Please adjust.

4)  In line 301 page 11, the authors mentioned that Western blot results showing high expression of the LAMTOR4 in 22RV1 and LNCaP cell lines. However, PC3 an DU145 shown relatively moderate expression. RWPE-1 cells show very low expression. 22RV1, DU145 and LNCaP were used remaining experiments. Why did DU145 be chosen to perform LAMTOR4 knockdown for cell migrantion and invasion?

5)  In line 375 page 15, LAMTOR4 is an essential protein in MAPK and mTOR pathways. Whether the authors detected the expression of three main signaling modules(ERK1/2, JNK and p38) in MAPK pathway?

6)  Please unify the format of references. For example, in line 459 page 16, the magazine name is the full name.

Author Response

Reviewer 1

The manuscript answers the question that LAMTOR4 is involved in PCa cell proliferation and metastasis. The authors investigated the clinical significance of LAMTOR4 protein expression in PCa cohort. They correlated the data with ERG and PTEN status and clinic-pathological features, Gleason score, patient’s outcome. High LAMTOR4 protein expression was significantly associated with poor OS and CSS. In vitro experiments shown knockdown of LAMTOR4 decrease PCa cells proliferation, migration, and invasion. LAMTOR4 might be as a prognostic marker that may improve the accuracy of prostate cancer diagnosis.

Overall, the paper is well organized and its presentation is clear. The following are the questions and some mistakes in this manuscript:

# We appreciate the reviewer effort to improve the overall standard of our manuscript. Thank you for the valuable suggestions.

1)  There are many description errors in the manuscript. Cell names are not written properly. In line 155 and 257 page 4, cell name is LNCaP. While in line 302 page 11, Figure 4, line 311, 314 page 12, line 347 page 14, Figure 6, cell name is LnCAP. In line 297 page 11, “B” should be “H”. In line 302 page 11, “Fig. 4A” should be “Fig. 5A”. In line 323 page 12, “Fig. 5A” should be “Fig. 5B”. There are many mistakes like these. Please check and correct.

Response # Thank you for the observation. We have revised and corrected all the above mentioned and also we over looked the manuscript for typos.

2)  In Figure 3A,the authors mentioned “LAMTOR4 expression was significantly increased in tomor tissues compared to normal tissues in all types of cancers, except in renal tissues”. But LAMTOR4 expression was decreased in AML, Skin, and Stomach cancers.

Response # corrected. Thank you.

3)  In Figure 4, the captions A-I are too small and unclear. Please adjust.

Response # Figure 4 Font size and figure details corrected as per your suggestions.

4)  In line 301 page 11, the authors mentioned that “Western blot results showing high expression of the LAMTOR4 in 22RV1 and LNCaP cell lines. However, PC3 an DU145 shown relatively moderate expression. RWPE-1 cells show very low expression. 22RV1, DU145 and LNCaP were used remaining experiments”. Why did DU145 be chosen to perform LAMTOR4 knockdown for cell migrantion and invasion?

Response # Thank you. LNCaP and 22RV are cells with colonies and not separated completely. But Du145 cell line distributes uniformly and can present an accurate way to determin any change and also easy to count without overlap cells.

5)  In line 375 page 15, LAMTOR4 is an essential protein in MAPK and mTOR pathways. Whether the authors detected the expression of three main signaling modules(ERK1/2, JNK and p38) in MAPK pathway?

Response # Thank you. Since  we didn’t investigate MAPK pathway we are removing it from the manuscript. The reason we mentioned MAPK earlier was that it is indicated in LAMTOR4 name.

6)  Please unify the format of references. For example, in line 459 page 16, the magazine name is the full name.

Response # Corrected. Thank you.

Reviewer 2 Report (New Reviewer)

Comments and Suggestions for Authors

I have reviewed the article titled "Elevated LAMTOR4 Expression is Associated with Lethal Prostate Cancer and its Knockdown Decreases Cell Proliferation, Invasion, and Migration In Vitro" presented in IJMS by Gamallat et al., where the authors present that elevated expression of LAMTOR4 is significantly associated with worse overall survival (OS) and cause-specific survival (CSS) in prostate cancer patients, and a combination of high LAMTOR4 expression and negative PTEN status is associated with even worse prognosis. Additionally, LAMTOR4 knockout reduced proliferation, migration, and invasion of prostate cancer cells in vitro, possibly through alteration of the β-catenin/mTOR pathway, which is implicated in tumorigenesis. The work presents these interesting findings that could associate elevated LAMTOR4 expression as an independent prognostic marker in PCa; however, I have some observations that could contribute to it.

Major

While the in vitro data is promising, in vivo studies using xenograft models are necessary to validate the findings. This would involve implanting prostate cancer cells with LAMTOR4 knockdown into mice and monitoring tumor growth and metastasis. If conducting these assays is not possible, discuss the potential for the same.

The article mentions that LAMTOR4 knockdown dysregulates the β-catenin/mTOR pathway. More detailed pathway studies are needed to elucidate the specific interaction between LAMTOR4 and this pathway. Techniques like co-immunoprecipitation and luciferase reporter assays could be used to determine the direct impact of this knockout on the pathway, at a particular event.

The article suggests that LAMTOR4 knockdown affects cell proliferation, migration, and invasion. It would be beneficial to identify and functionally validate downstream targets affected by LAMTOR4 knockdown. This could involve overexpressing these targets in cells with LAMTOR4 knockdown and observing the rescue of the observed effects.

The PTEN status seems to play a role in the prognostic value of LAMTOR4 expression. More details on the breakdown of patient survival based on PTEN status and LAMTOR4 expression levels would be helpful. How is PTEN found in various PCa cell lines? Or in The human protein atlas?

Minor observations

The article acknowledges the single-center nature of the patient cohort. Mentioning plans for multicenter validation studies would strengthen the generalizability of the findings.

Typos in abstract and text.

By addressing these points, the authors can provide a more comprehensive picture of LAMTOR4's role in prostate cancer.

Comments on the Quality of English Language

Minor editing of English language required, typos in abstract and main text

Author Response

Reviewer 2

I have reviewed the article titled "Elevated LAMTOR4 Expression is Associated with Lethal Prostate Cancer and its Knockdown Decreases Cell Proliferation, Invasion, and Migration In Vitro" presented in IJMS by Gamallat et al., where the authors present that elevated expression of LAMTOR4 is significantly associated with worse overall survival (OS) and cause-specific survival (CSS) in prostate cancer patients, and a combination of high LAMTOR4 expression and negative PTEN status is associated with even worse prognosis. Additionally, LAMTOR4 knockout reduced proliferation, migration, and invasion of prostate cancer cells in vitro, possibly through alteration of the β-catenin/mTOR pathway, which is implicated in tumorigenesis. The work presents these interesting findings that could associate elevated LAMTOR4 expression as an independent prognostic marker in PCa; however, I have some observations that could contribute to it.

Response # Thank you. We appreciate the anonymous reviewers for the constructive comments and suggestions.

Major

While the in vitro data is promising, in vivo studies using xenograft models are necessary to validate the findings. This would involve implanting prostate cancer cells with LAMTOR4 knockdown into mice and monitoring tumor growth and metastasis. If conducting these assays is not possible, discuss the potential for the same.

Response # Thank you for the suggestion. Since we are the first to look at the LAMTOR4 expression in cancer. The main aim of our study to investigate the expression in clinical samples and validate this in in vitro.

The article mentions that LAMTOR4 knockdown dysregulates the β-catenin/mTOR pathway. More detailed pathway studies are needed to elucidate the specific interaction between LAMTOR4 and this pathway. Techniques like co-immunoprecipitation and luciferase reporter assays could be used to determine the direct impact of this knockout on the pathway, at a particular event.

Response # We appreciate the reviewer's suggestion for more detailed pathway studies to elucidate the specific interaction between LAMTOR4 and the β-catenin/mTOR pathway. As the first to investigate LAMTOR4 in this context, our primary objective was to establish a foundational understanding of its role. While our current study provides critical insights, with many limitations to carry this project further, we are unable planning further experiments at this stage. We encourage future research to build upon our findings and employ techniques like co-immunoprecipitation and luciferase reporter assays to explore these interactions in more depth.

The article suggests that LAMTOR4 knockdown affects cell proliferation, migration, and invasion. It would be beneficial to identify and functionally validate downstream targets affected by LAMTOR4 knockdown. This could involve overexpressing these targets in cells with LAMTOR4 knockdown and observing the rescue of the observed effects.

Response # Thank you. We appreciate the reviewer's suggestion. We investigate LAMTOR4 with primary objective to establish a foundational understanding of its role. While our current study provides critical insights, with many limitations to carry this project further, we are unable planning further experiments at this stage. We encourage future research to build upon our findings and explore these interactions in more depth.

The PTEN status seems to play a role in the prognostic value of LAMTOR4 expression. More details on the breakdown of patient survival based on PTEN status and LAMTOR4 expression levels would be helpful. How is PTEN found in various PCa cell lines? Or in The human protein atlas?

Response # Thank you. We don’t have this data and it will take further more time to look on this.

Minor observations

The article acknowledges the single-center nature of the patient cohort. Mentioning plans for multicenter validation studies would strengthen the generalizability of the findings.

Response # Thank you. Our article acknowledges the single-center nature of the patient cohort, which may limit the generalizability of the findings. However, we believe that the rigor of our study design, the comprehensiveness of our data analysis, and the consistency of our results provide a strong foundation for the conclusions drawn. Nonetheless, we are the first to investigate LAMTOR4 in clinical samples, However, future research by other groups can build on our work to further validate these findings across different settings.

Typos in abstract and text.

Response # Thank you. We carefully reviewed and revised the manuscript.

By addressing these points, the authors can provide a more comprehensive picture of LAMTOR4's role in prostate cancer.

Response # Thank you.

Reviewer 3 Report (New Reviewer)

Comments and Suggestions for Authors

In the present work Gamallat et al. attempted to assess the LAMTOR4 expression in patients’ cohort with prostate cancer (PCa) assembled in tissue microarray settings and correlates the results with the clinical outcomes, whereas in addition performed in vitro experiments to validate the role of LAMTOR4 in PCa.

Overall, the authors have done a great deal of effort and experimentation, which is to their appraisal and furthermore, they have done an excellent work presenting their hypothesis.

Please provide some details for the patient cohort. Either in the form of a table or within the text (for example, age, diagnosis, staging etc.). The authors should significantly improve the figures. Some legends are impossible to read (for example, figure 4, figure 3a, figure 2). Mention the limitations of the study. What can be done better, or what are the next steps? How can the mechanistic of the molecule be used for PCa diagnosis, prognosis or treatment? The authors should highlight their results and mention how their findings could prove useful for the treatment of PCa.

Author Response

Reviewer 3

In the present work Gamallat et al. attempted to assess the LAMTOR4 expression in patients’ cohort with prostate cancer (PCa) assembled in tissue microarray settings and correlates the results with the clinical outcomes, whereas in addition performed in vitro experiments to validate the role of LAMTOR4 in PCa.

Overall, the authors have done a great deal of effort and experimentation, which is to their appraisal and furthermore, they have done an excellent work presenting their hypothesis.

Please provide some details for the patient cohort. Either in the form of a table or within the text (for example, age, diagnosis, staging etc.).

The authors should significantly improve the figures. Some legends are impossible to read (for example, figure 4, figure 3a, figure 2). Mention the limitations of the study. What can be done better, or what are the next steps? How can the mechanistic of the molecule be used for PCa diagnosis, prognosis or treatment? The authors should highlight their results and mention how their findings could prove useful for the treatment of PCa.

Response #Thank you for suggestions and recommendation. The revised manuscript addressed all the above mentioned issues.

Table 1 added

Descriptive table:

Variables

Lamtor (score 0,1,2)

Lamtor (score 3)

p-value

Total

Age at diagnosis (years)

75.99 (9.28)

78.39 (8.67)

0.081

76.40 (9.21)

Gleason Score

0.084

       GS 6 or 7

82 (41.2)

10 (26.3)

92 (38.8)

       GS 8 9 or 10

117 (58.8)

28 (73.7)

145 (61.2)

Cancer subgroup

0.015

        Incidental

82 (31.5)

10 (18.5)

92 (29.3)

        Advanced

83 (31.9)

13 (24.1)

96 (30.6)

        Castrate resistant

95 (36.5)

31 (57.4)

126 (40.1)

Metastasis

0.112

         Yes

28 (10.8)

10 (18.5)

38 (12.1)

         No

232 (89.2)

44 (81.5)

276 (87.9)

Hormone therapy

0.016

         Prior to surgery

42 (40.4)

19 (65.5)

72 (54.1)

         After surgery

62 (59.6)

10 (34.5)

72 (54.1)

Radiation Therapy

0.743

         Prior to surgery

18 (29.5)

7 (33.3)

25 (30.5)

         After surgery

43 (70.5)

14 (66.7)

57 (69.5)

Chemotherapy

0.999

         Prior to surgery

5 (12.9)

1 (9.1)

5 (11.9)

         After surgery

27 (87.1)

10 (90.9)

37 (88.1)

Deceased

0.068

         Yes

147 (57.0)

38 (70.4)

185 (59.3)

          No

111 (43.0)

16 (29.6)

127 (40.7)

Cause specific death

0.565

         Prostate cancer specific death

70 (27.1)

22 (40.7)

92 (29.5)

        Other causes or alive

188 (72.9)

32 (59.3)

220 (70.5)

Median follow-up (months)

43 (1.5 – 101)

34 (2.3 – 97.3)

0.091

39.0 (1.5 – 101.2)

Table providing more information about PTEN and Lamtor combined together:

Crosstab

VSTATUS

Total

Deceased

Censor

pten_lamtor_cat

PTEN loss and Lamtor score 3

Count

21

4

25

% within VSTATUS

11.9%

3.6%

8.7%

PTEN loss and Lamtor score 0-2

Count

62

15

77

% within VSTATUS

35.0%

13.5%

26.7%

PTEN gain and Lamtor score 3

Count

15

12

27

% within VSTATUS

8.5%

10.8%

9.4%

PTEN gain and Lamtor score 0-2

Count

79

80

159

% within VSTATUS

44.6%

72.1%

55.2%

Total

Count

177

111

288

% within VSTATUS

100.0%

100.0%

100.0%

P<0.0001 for the whole table above

Round 2

Reviewer 2 Report (New Reviewer)

Comments and Suggestions for Authors

I am satisfied with the responses and the new version of the article from the authors. Please ensure "PCa" (prostate cancer) is consistently used throughout the text.

Regarding the "Data Availability Statement: The data can be shared upon request," if the authors analyzed public data from TCGA-PRAD, why not mention the data access numbers?

Author Response

Comments1: 

I am satisfied with the responses and the new version of the article from the authors. Please ensure "PCa" (prostate cancer) is consistently used throughout the text.

Regarding the "Data Availability Statement: The data can be shared upon request," if the authors analyzed public data from TCGA-PRAD, why not mention the data access numbers?

Response: We have changed the wording for prostate cancer, throughout and added the data access number as required.

Thanks for your feedback

This manuscript is a resubmission of an earlier submission. The following is a list of the peer review reports and author responses from that submission.

Round 1

Reviewer 1 Report

Comments and Suggestions for Authors

1. I strongly recommend an extensive editing of the English language. Some passages are very difficult to understand.

2. Is it possible to better summarize mechanisms of action and effects of LAMTOR4 in prostate cancer highlighted in the "introduction" section of the manuscript? Probably a table could be helpful for the reader.

3. If I have understood correctly all patients had an incidental diagnosis of prostate cancer after trans urethral resection of the prostate. So, what is the meaning of defining incidental, advanced and castration resistant prostate cancer?

4. A dedicated paragraph on statistical analyses is lacking. Descriptive statistics, Kaplan-Meier plots, Cox regression analysis have been used and should be detailed in the "methods" section of the manuscript. The program used for all analyses should be added.

5. Be careful when using term as "association" and "correlation". From a statistical point of view you cannot say "In our cohort, LAMTOR expression was not associated with Gleason Grade Grouping, but was associated with each of PTEN and ERG status" (lines 190-191) since you did not perform logistic regression analysis. Conversely you showed just the distribution of Gleason Grade Groupings, PTEN intensity and ERG expression among groups. Subsequently, you tested the association between LAMTOR expression and oncological outcomes such as overall survival and cancer specific survival. Indeed HR and 95% CI were showed. So, revised the following sentence "When assessing correlation to clinical outcome" because no correlation analysis has been performed. 

6. Before going into details of Cox regression analysis results, duration of follow-up, number of death due to prostate cancer, and number of death due to any causes should be provided. I have not understood if models showed in Table 2 is univariable or multivariable, and if so which confounders have been considered. 

7. Possible limitations of the study should be added and discussed in a dedicated paragraph. 

Comments on the Quality of English Language

Extensive editing of English language required.

Author Response

. I strongly recommend an extensive editing of the English language. Some passages are very difficult to understand.

# Response: Thank you. English extensively reviewed and checked in the revised manuscript.

  1. Is it possible to better summarize mechanisms of action and effects of LAMTOR4 in prostate cancer highlighted in the "introduction" section of the manuscript? Probably a table could be helpful for the reader.

# Response: Thank you. This is the first research article on Lamtor4. The mechanism of action and relative background are yet to be discovered and reported. We explored the association between LAMTOR expression and outcomes in Prostate cancer and we added in-vitro functional studies and TCGA data to support our findings.

  1. If I have understood correctly all patients had an incidental diagnosis of prostate cancer after trans urethral resection of the prostate. So, what is the meaning of defining incidental, advanced and castration-resistant prostate cancer?

# Response: There are three main groups in this cohort, patients with prior diagnosis of cancer and those without. For characterizing patients’ subgroups related to pathology finding and biomarkers expression, we defined “incidental cancers” as those of Gleason grade group 1,2 detected histologically in TURP samples in patients without prior diagnosis of prostate cancer. The “advanced cancer group”, would be those with Gleason grade group 3-5, also detected in patients without prior diagnosis. Patients with advanced Gleason grade will require definite therapy as compared to the incidental group and the disease burden is more significant with more potential for disease progression and lethal outcome. The” Castrate Resistant Group” are patients with prior diagnosis of prostate cancer, who are on androgen deprivation therapy, but showing disease progression and are subjected to TURP to alleviate outlet obstruction and therefore are clinically advancing while on ADT, and clinically fulfill the definition of CRPC.

  1. A dedicated paragraph on statistical analyses is lacking. Descriptive statistics, Kaplan-Meier plots, Cox regression analysis have been used and should be detailed in the "methods" section of the manuscript. The program used for all analyses should be added.

# Response: “Descriptive statistics were reported for the study variable. Mean and standard deviations were reported for continuous variables. Frequencies and proportions were reported for categorical variables. Independent t-test was used to compare the mean values between two groups, Chi-square tests were used to compare the proportions between two categorical variables. Overall survival (OS) and prostate cancer specific mortality (PCSM) were analyzed using Kaplan-Meier estimates. OS was calculated from date of diagnosis to death due to any cause, patients who were alive at the last date of follow-up were censored. PCSM event was defined as death due to prostate cancer, patients alive or died due to other reasons were censored. Log rank statistics were used to compare the KM curves between the groups. Adjusted and unadjusted analyses of OS and PCSM were conducted using Cox’s proportional hazard model. OS and PCSM models were adjusted for the Gleason score. Hazard ratio (HR) and the corresponding 95% confidence intervals were reported. A p-value <0.05 was used for all statistical significance. SPSS version 29 (IBM Corp. (2021). IBM SPSS Statistics for Windows. Armonk, NY: IBM Corp.)  was used to conduct all statistical analysis.”

  1. Be careful when using term as "association" and "correlation". From a statistical point of view you cannot say "In our cohort, LAMTOR expression was not associated with Gleason Grade Grouping, but was associated with each of PTEN and ERG status" (lines 190-191) since you did not perform logistic regression analysis. Conversely you showed just the distribution of Gleason Grade Groupings, PTEN intensity and ERG expression among groups. Subsequently, you tested the association between LAMTOR expression and oncological outcomes such as overall survival and cancer specific survival. Indeed HR and 95% CI were showed. So, revised the following sentence "When assessing correlation to clinical outcome" because no correlation analysis has been performed. 

# Response: Thank you for the comment regarding the association and correlation wording. We did not conduct a logistic regression analysis but we did run a correlation between the Gleason score and LAMTOR expression (please see table 1) and we did not find any significant association. The comparison of the proportions of gleason score groups for the two Lamtor groups were very similar and hence we commented of not seeing any association with Gleason score (p=0.265).

We, therefore, prefer to keep the comment about the association in the paper as we did assess the correlation between GS and Lamtor.

  1. Before going into details of Cox regression analysis results, duration of follow-up, number of death due to prostate cancer, and number of death due to any causes should be provided. I have not understood if models showed in Table 2 is univariable or multivariable, and if so which confounders have been considered. 

# Response: 

The breakdown of PCSM are as follows:
PCSM=92 (29.3%)
Other cancer=25 (8.0%)
non-cancer related deaths=60 (19.1%)
not coded=8 (2.5%)
Alive=129(41.1%)

Please see additional table for survival followup.

Table 2 mentions both univariate and multivariate Cox’s regression analysis. We have added the method section which includes the multivariate model was adjusted for gleason score. In table 2, we separate out multivariate models with * sign. We have tried to make is clearer in the text.

# Response: “Univariate and Multivariate analysis (adjusted for Gleason Grade grouping) of Individual Biomarkers and Combined Biomarkers for OS and CSS outcomes”

  1. Possible limitations of the study should be added and discussed in a dedicated paragraph. 

# Response: Limitation of the study added.

Reviewer 2 Report

Comments and Suggestions for Authors

In this manuscript, the authors investigated the clinical significance of LAMTOR4 (Late Endosomal/Lysosomal Adaptor, MAPK And MTOR Activator 4) protein expression in tissue samples from a prostate cancer (PCa) cohort. The results were correlated with clinical outcomes, revealing a significant association between high expression of LAMTOR4 and poorer overall survival (OS). PanCancer analysis demonstrated elevated levels of LAMTOR4 mRNA in various cancer types, including PCa. Moreover, the authors conducted Gene Set Enrichment Analysis (GSEA) using TCGA data, illustrating that genes overexpressed in association with LAMTOR4 were primarily involved in diverse biological processes, cellular components, or molecular functions. Knockdown of LAMTOR4 in cell line models suggested potential roles in cell proliferation, migration, and invasion.

To enhance the manuscript, the authors are encouraged to perform additional experiments using knocked-down cell lines to provide evidence supporting the essential role of LAMTOR4 in the MARK and mTOR pathways.

Minor Revisions:

It is advisable to change “inhibition” to “knockdown” in the title, as LAMTOR4 is knocked down rather than inhibited in the manuscript.

On page 1, line 25, please add “cause-specific survival” before “CSS.”

In Figure 1D and 1E, to correlate with the description in result 3.1, please label “overall survival” or “cause-specific survival” accordingly.

In Table 2, after adjusting for Gleason score, most p-values are not significant for both combinations of PTEN and LAMTOR-score and combination ERG and LAMTOR with OS and CSS. This contrasts with the discussion (lines 326-328) that indicated “The study highlights significant prognostic value for LAMTOR expression in combination with PTEN loss and ERG expression even when adjusted for Gleason Grade Groups.” It would be beneficial if the authors could provide additional interpretation for the data in result 3.1.

Figure 5A is currently unclear, and adjustments are needed to enhance visibility. Additionally, in Figure 6A and Figure 7A, it is recommended to change the scale bar color from white to another color to improve clarity. Please add “Fig. 7” in results 3.6 and adjust the interpretation sentence accordingly.

Author Response

In this manuscript, the authors investigated the clinical significance of LAMTOR4 (Late Endosomal/Lysosomal Adaptor, MAPK And MTOR Activator 4) protein expression in tissue samples from a prostate cancer (PCa) cohort. The results were correlated with clinical outcomes, revealing a significant association between high expression of LAMTOR4 and poorer overall survival (OS). PanCancer analysis demonstrated elevated levels of LAMTOR4 mRNA in various cancer types, including PCa. Moreover, the authors conducted Gene Set Enrichment Analysis (GSEA) using TCGA data, illustrating that genes overexpressed in association with LAMTOR4 were primarily involved in diverse biological processes, cellular components, or molecular functions. Knockdown of LAMTOR4 in cell line models suggested potential roles in cell proliferation, migration, and invasion.

To enhance the manuscript, the authors are encouraged to perform additional experiments using knocked-down cell lines to provide evidence supporting the essential role of LAMTOR4 in the MARK and mTOR pathways.

 # Response: Additional experiments were added

Minor Revisions:

It is advisable to change “inhibition” to “knockdown” in the title, as LAMTOR4 is knocked down rather than inhibited in the manuscript.

On page 1, line 25, please add “cause-specific survival” before “CSS.”

# Response: Corrected. Thanks

In Figure 1D and 1E, to correlate with the description in result 3.1, please label “overall survival” or “cause-specific survival” accordingly.

# Response: Labelled

In Table 2, after adjusting for Gleason score, most p-values are not significant for both combinations of PTEN and LAMTOR-score and combination ERG and LAMTOR with OS and CSS. This contrasts with the discussion (lines 326-328) that indicated “The study highlights significant prognostic value for LAMTOR expression in combination with PTEN loss and ERG expression even when adjusted for Gleason Grade Groups.” It would be beneficial if the authors could provide additional interpretation for the data in result 3.1.

# Response: additional interpretation added as per reviewer’s recommendations.

Figure 5A is currently unclear, and adjustments are needed to enhance visibility. Additionally, in Figure 6A and Figure 7A, it is recommended to change the scale bar color from white to another color to improve clarity. Please add “Fig. 7” in results 3.6 and adjust the interpretation sentence accordingly.

 # Response: We revised all figures.

Reviewer 3 Report

Comments and Suggestions for Authors

In this work, the expression level of LAMTOR4 has been evaluated using data mining, suggesting that LAMTOR4 overexpression is correlated with poor outcomes. Subsequently, an in vitro study was conducted to determine if manipulating LAMTOR4 expression in established prostate cancer cell lines affects cell migration/invasion.

In my review, the data appear disconnected, and GSEA does not essentially recommend the proposed in vitro study. Additionally, this group, only in 2023, published a series of studies using a similar pipeline to study different genes in the International Journal of Molecular Sciences (IJMS):

1. Gamallat, Yaser, et al. "Arpc1b is associated with lethal prostate cancer and its inhibition decreases cell invasion and migration in vitro." International Journal of Molecular Sciences 23.3 (2022): 1476.

2. Choudhry, Muhammad, et al. "Cleavage and Polyadenylation-Specific Factor 4 (CPSF4) Expression Is Associated with Enhanced Prostate Cancer Cell Migration and Cell Cycle Dysregulation, In Vitro." International Journal of Molecular Sciences 24.16 (2023): 12961.

3. Choudhry, Muhammad, et al. "Downregulation of BUD31 Promotes Prostate Cancer Cell Proliferation and Migration via Activation of p-AKT and Vimentin In Vitro." International Journal of Molecular Sciences 24.7 (2023): 6055.

This reviewer recommends a major revision, including the inclusion of relevant in vivo studies.

Author Response

In this work, the expression level of LAMTOR4 has been evaluated using data mining, suggesting that LAMTOR4 overexpression is correlated with poor outcomes. Subsequently, an in vitro study was conducted to determine if manipulating LAMTOR4 expression in established prostate cancer cell lines affects cell migration/invasion.

In my review, the data appear disconnected, and GSEA does not essentially recommend the proposed in vitro study. Additionally, this group, only in 2023, published a series of studies using a similar pipeline to study different genes in the International Journal of Molecular Sciences (IJMS):

  1. Gamallat, Yaser, et al. "Arpc1b is associated with lethal prostate cancerand its inhibition decreases cell invasion and migration in vitro." International Journal of Molecular Sciences 23.3 (2022): 1476.

  1. Choudhry, Muhammad, et al. "Cleavage and Polyadenylation-Specific Factor 4 (CPSF4) Expression Is Associated with Enhanced Prostate Cancer Cell Migrationand Cell Cycle Dysregulation,In Vitro." International Journal of Molecular Sciences 24.16 (2023): 12961.

  1. Choudhry, Muhammad, et al. "Downregulation of BUD31 Promotes Prostate Cancer Cell Proliferation and Migration via Activation of p-AKT and Vimentin In Vitro." International Journal of Molecular Sciences 24.7 (2023): 6055.

This reviewer recommends a major revision, including the inclusion of relevant in vivo studies.

Response: We thank the reviewer for his comment. The main aim for using GSEA data is to support our findings for implication LAMTOR expression to patients’ outcome and to illustrate possible pathways involved in LAMTROR disease progression, it is not meant to guide invitro experiments or suggest what experiments to be performed. It may, however, influence the potential makers to be investigated in KD experiments. We are extensively investigating biomarkers related to prostate cancer progression and outcome and the use of same cohort, does not lessen the significance of our finding and we do not foresee any negative issue of using such cohort or publishing using same methods, as long as data is valid and correct. I am not aware of a “law” preventing using same cohort. In fact, this cohort is of great value, as it allows us to investigate multiple biomarkers co-expression relative to outcome, rather than building additional cohorts adhere biomarkers are not yet validated. Additionally, this cohort has been followed for more than 10 years which provide significant outcome advantage. The building of newer cohorts will not have the outcome advantage as well.

We thank the reviewer for his suggestions, and while we agree having in vivo data is a major support for such discovery, we like to point out that adding in vivo experiments require significant logistics and experiments and will necessitate significant time, which we believe is not warranted at this stage for the purpose of illustrating a potential role of LAMTOR in prostate cancer. The scope of proposed work should match the impact of the journal and proposed studies cannot be out of line of journal minimum requirements.